# Hypervolume Niche Dynamics and Global Invasion Risk of *Phenacoccus solenopsis* under Climate Change

**DOI:** 10.3390/insects15040250

**Published:** 2024-04-05

**Authors:** Shaopeng Cui, Huisheng Zhang, Lirui Liu, Weiwei Lyu, Lin Xu, Zhiwei Zhang, Youzhi Han

**Affiliations:** 1College of Forestry, Shanxi Agricultural University, Jinzhong 030801, China; shaopengcui@126.com (S.C.); zhiweizhang2012@163.com (Z.Z.); 2Shanxi Dangerous Forest Pest Inspection and Identification Center, Jinzhong 030801, China

**Keywords:** hypervolume niche, niche shift, *Phenacoccus solenopsis*, MaxEnt model, climate change

## Abstract

**Simple Summary:**

The cotton mealybug, *Phenacoccus solenopsis*, native to the USA, is a globally invasive quarantine pest. This pest has now spread to more than 43 countries and covered all continents except Antarctica, posing a serious threat to agricultural and forestry production, as well as biosecurity. The niche conservatism hypothesis states a tendency for species to maintain their original ecological niches in time and space and has been widely debated. In this study, based on the multidimensional hypervolume analysis and species distribution model, we identified the niche dynamics of this pest and further assessed its global invasion risk. The results indicated that the niche hypervolume in invasive ranges was significantly larger than that in native ranges, with 99.45% of the niche differentiation contributed by niche expansion rather than spatial replacement. Niche expansion was especially evident in Oceania and Eurasia. Compared to 2017, the current area of suitable habitats had increased significantly and this pest would expand northwards from the south in future climate change scenarios. The invasion risk in North Africa, northern China, areas along the Mediterranean, and northern Europe should not be ignored.

**Abstract:**

As a globally invasive quarantine pest, the cotton mealybug, *Phenacoccus solenopsis*, is spreading rapidly, posing serious threats against agricultural and forestry production and biosecurity. In recent years, the niche conservatism hypothesis has been widely debated, which is particularly evident in invasive biology research. Identifying the niche dynamics of *P. solenopsis*, as well as assessing its global invasion risk, is of both theoretical and practical importance. Based on 462 occurrence points and 19 bioclimatic variables, we used *n*-dimensional hypervolume analysis to quantify the multidimensional climatic niche of this pest in both its native and invasive ranges. We examined niche conservatism and further optimized the MaxEnt model parameters to predict the global invasion risk of *P. solenopsis* under both current and future climate conditions. Our findings indicated that the niche hypervolume of this pest in invasive ranges was significantly larger than that in its native ranges, with 99.45% of the niche differentiation contributed by niche expansion, with the remaining less than 1% explained by space replacement. Niche expansion was most evident in Oceania and Eurasia. The area under the receiver operating characteristic curve (0.83) and true skill statistic (0.62) indicated the model’s robust performance. The areas of suitable habitats for *P. solenopsis* are increasing significantly and the northward spread is obvious in future climate change scenarios. North Africa, northern China, Mediterranean regions, and northern Europe had an increased invasion risk of *P. solenopsis*. This study provided scientific support for the early warning and control of *P. solenopsis.*

## 1. Introduction

The elucidation of species distribution patterns, as well as the development of biodiversity management techniques, hinge on the understanding of ecological niches. Niche conservatism, a tendency for species to maintain their original ecological niches in time and space, is generally hypothesized and accepted [1]. Recently, several studies have reported species’ niches changing, in terms of breadth and location (i.e., niche shifts), causing a significant discourse on niche conservatism, which is particularly evident in invasive biology research [2,3], where an alien species is considered invasive because it dominates the invaded community and produces a harmful impact. For example, by comparing the climatic niches between native and invasive ranges, researchers found that significant niche shifts occurred for 12 out of 22 globally invasive insects [4]. Similarly, Atwater and Barney [5] suggested that climatic niche shifts in 815 introduced plant species affected their predicted distributions. However, an analysis of 434 invasive species from 86 studies indicated that most invasive species maintained climate fidelity, supporting the niche conservatism hypothesis [1].

There are currently four primary approaches for analyzing niche dynamics, as follows: the ordination approach, the univariate approach, the ecological niche model approach, and the multidimensional hypervolume approach [1]. Among these, the ordination and hypervolume approaches project the variables related to species distribution onto the two-dimensional and multidimensional space, respectively, allowing for a direct comparison of niche differences; these approaches are, thus, considered to be the most effective methods [1,3,6]. However, the hypervolume approach is rarely used, mainly due to a lack of quantification methods [1,7]. In the 1950s, Hutchinson [8] introduced the concept of the *n*-dimensional niche hypervolume, defining a multidimensional space for each species by treating each condition that affects the species’ survival or each resource that can be used by the species as a dimension. Despite the intuitiveness of the concept, describing and characterizing the niche hypervolume of a dataset has remained challenging for investigators [1,6]. Researchers have recently proposed a novel algorithm that can describe the shape and density of an *n*-dimensional hypervolume, which has been iteratively refined [7,9]. This method is now widely used in ecology and evolutionary biology research and has become a frontier and focus of multidimensional niche research (e.g., [10,11]).

The species distribution model (SDM), which quantifies the relationship between distribution data and environmental factors to predict the potential distribution of a species, is commonly used to assess the invasion risk of alien species [12,13]. Niche conservatism is a key assumption of this model, which means that species occupy similar environmental niches in both invasive and native ranges. More than ten SDM algorithms have been developed, such as MaxEnt, GLM, RF, BIOCLIM, etc., among which, MaxEnt is the most representative [14]. By comparing 16 SDMs, Elith et al. [15] found that MaxEnt provides the most accurate predictions compared to the other models. In addition, MaxEnt showed good performances even with a small occurrence dataset [16]. As a result, MaxEnt is widely used in invasive biology research. However, using the default parameters of MaxEnt without consideration of species niche conservatism and model complexity would result in unreliable predictions [2,17,18].

The cotton mealybug, *Phenacoccus solenopsis* Tinsley (Hemiptera: Pseudococcidae), is native to the USA [19] and has, subsequently, been found in several countries in succession [20], including Mexico, Brazil, Chile, Australia, Malaysia, Nigeria, etc. Currently, this pest has been reported across all continents except Antarctica, with more than 200 species of host plants recorded [21]. *P. solenopsis* completes at least 12 generations per year [21,22]. Its adult females and nymphs generally cluster in the tender parts of host plants, affecting plant growth by sucking sap. *P. solenopsis* also secrete honeydew to attract red fire ants, *Solenopsis invcta*, reducing the probability of their own predation and parasitism [23]. As a result, this pest has caused significant economic losses in numerous countries (especially in Asia) [21,24]. From 2000 to 2005, *P. solenopsis* was recorded in South American countries, such as Brazil [25], Chile [26], and Argentina [27]. After 2005, *P. solenopsis* invaded Pakistan and India in Asia [28,29], UK in Europe [21], and then Australia [30]. This pest was first reported in Nigeria, Africa [31] and Guangdong province, China in 2008. In 2010, *P. solenopsis* spread to nine provinces in China in just one year and was quickly classified as a quarantine pest in agriculture and forestry. Currently, *P. solenopsis* has spread to more than 43 countries [21], posing a serious threat to agricultural and forestry production, as well as biosecurity. Therefore, assessing the niche shifts of this pest, elucidating the applicability of SDMs to predict the potential distribution, and further assessing its global invasion risk are important for its control and management.

In this study, we first organized the distribution records of *P. solenopsis* and integrated the climatic variables through principal component analysis (PCA). Secondly, we used the hypervolume approach to quantify the *n*-dimensional niche hypervolumes of *P. solenopsis* in both its native and invasion ranges and examined its niche conservatism. Finally, the MaxEnt model parameters were optimized to identify key climatic variables affecting *P. solenopsis* distribution. We then predicted the species’ global invasion risk and examined habitat changes in various climate change scenarios, aiding scientific prevention and control efforts.

## 2. Materials and Methods

### 2.1. Data Sources

#### 2.1.1. Species Occurrence Data

A literature review found that Wei et al. [12] summed up the global distribution records of the *P. solenopsis* dated up to 2017, with a total of 201 distribution points after quality control. Wang [22] surveyed and sampled 198 points in 31 provinces in China and reported the first invasion of *P. solenopsis* in Shandong province, using the molecular identification of the mitochondrial COI gene. Based on the above findings, Zhang et al. [32] updated the worldwide distribution of *P. solenopsis* since 2017 by reviewing the relevant literature published in the past five years [33,34,35,36,37,38,39,40,41,42] and searching the Global Biodiversity Information Facility (GBIF) and Centre for Agriculture and Bioscience International (CABI) databases, obtaining 706 distribution points. This study directly used these distribution data for simulation and analysis.

To eliminate the impacts of spatial autocorrelation and sampling bias, we used SDMtoolbox 2.4 to create 5 km × 5 km grids for sparsity processing and randomly deleted redundant points within the grid to ensure that each grid contained, at most, only one record [12]. In the end, a total of 462 occurrence records of *P. solenopsis* were retained, including 60 native range records and 402 invasive range records.

#### 2.1.2. Bioclimatic Variables

Both current and future bioclimatic data were downloaded from the WorldClim database (version 2.1, http://www.worldclim.org, accessed on 30 March 2024), including 19 bioclimatic variables, at a spatial resolution of 2.5′ (5 km × 5 km). These data have been widely used for niche analysis and the prediction of suitable habitats for invasive insects [43].

As in the sixth phase of the Coupled Model Intercomparison Project (CMIP6), we selected the Beijing Climate Center Climate System Model 2 Medium Resolution (BCC-CSM2-MR) to model climate changes in the late 21st century (years 2081–2100). Following the shared socio-economic pathways (SSPs), two greenhouse gas emission scenarios were selected, namely the low radiative forcing scenario (SSP1-2.6, 2.6 W·m^−2^) and the highest forcing scenario in the absence of climate policy interference (SSP5-8.5, 8.5 W·m^−2^).

Multicollinearity often exists between environmental variables and seriously affects the accuracy of model prediction. However, the arbitrary removal of relevant variables may result in a loss of useful environmental information. Therefore, we used R (version 4.3.1 [44]) to perform PCA on the 19 bioclimatic variables and calculated the correlation coefficient matrix to analyze the correlation between the variables. Two variables were considered correlated when *|r|* ≥ 0.75. Finally, we retained principal components (PCs) with eigenvalue >1 (Kaiser–Guttman criterion [45]), which explained >99.5% of the cumulative proportion of variation, to obtain orthogonal predictor variables for analyzing the niche dynamics and invasion potential of *P. solenopsis*.

### 2.2. Niche Estimation

The quantification framework of niche dynamics proposed by Petitpierre et al. [6] is currently the most widely used technique in niche conservatism analyses. However, recent findings have shown that this technique, when compared to the multidimensional hypervolume approach, has a risk of information loss that may affect its accuracy [46]. Therefore, this study used the hypervolume method proposed by Blonder et al. [9] to construct an *n*-dimensional niche hypervolume to quantify the multidimensional climatic niche of *P. solenopsis*. Using Gaussian kernel density estimation, hypervolumes were generated in R using the hypervolume package [47] and the bandwidth of each variable axis was optimized through cross-validation. First, two hypervolumes were constructed to depict population niches in both the native and invasive ranges, respectively, and then the invasive range was subdivided into four major regions, namely Eurasia, Africa, Oceania, and South America [12]. The hypervolumes of the four regions were generated to finally obtain a total of six hypervolumes and their hypervolume volumes (HVs) were calculated, respectively, to characterize the niche size of the ecological niche.

To assess the niche shifts during *P. solenopsis* invasion, the differences between hypervolumes were explored in relation to the following two aspects: distance and similarity [11]. Each hypervolume was projected onto orthogonal axes consisting of climatic variable pairs using the hypervolume_distance function. Then, the Euclidean distance between centroids was calculated to characterize the spatial distance between the hypervolumes [9]. The kernel.beta function from the BAT package estimated the total beta diversity (β_total_) [11], for measuring the similarity (i.e., the degree of overlap) between two hypervolumes. β_total_ was then further decomposed into two potential processes, as follows: space replacement between hypervolumes (β_replacement_), which reflects the niche shift due to the change in spatial location, and the net difference of niche space enclosed by each hypervolume (β_richness_), which reflects niche expansion or contraction [10]. The value of β_total_ ranges from 0 (identical niches) to 1 (completely different niches) and β_total_ = β_replacement_ + β_richness_.

### 2.3. MaxEnt Modeling

#### 2.3.1. Model Optimization

The MaxEnt model (version 3.4.4 [16]) was used to predict the potential geographic distribution of *P. solenopsis*. Compared with other species distribution models, MaxEnt delivers an outstanding accuracy with presence-only data and has been widely used for the analysis and prediction of suitable habitats for invasive alien species [14,15].

The feature combination (FC) and regularization multiplier (RM) were optimized using the ENMeval 2.0 R package [48] to reduce the model complexity and select the optimal combination for modeling. Block partition was used to divide the record points into four equal sets, three for training and one for testing. The RM was set to 0.5 to 4, at an interval of 0.5, for a total of 8 values. The model provides five features, namely linear (L), quadratic (Q), product (P), threshold (T), and hinge (H), and the FCs used in this study were L, H, LQ, LQH, LQHP, and LQHPT. The above 48 parameter combinations were screened to assess the model complexity and goodness-of-fit based on the corrected Akaike information criterion (AICc) [49]. The model was determined as credible when ΔAICc < 2 [50]. Additionally, the area under the receiver operating characteristic curve (AUC) is a comprehensive indicator to assess the sensitivity and specificity of the model, so the difference between training and calibration (AUC_diff_) was used to measure the degree of model overfitting and to, finally, select the optimal parameter combination [18,48].

#### 2.3.2. Model Establishment

Based on the optimal combination of parameters, we used 10-fold cross-validation to test the model performance. The maximum number of iterations was set to 5000. We assessed the importance of climatic variables using the Jackknife procedure.

Different thresholding methods can lead to great differences in prediction results but, to date, there is no generally accepted method for calculating the most suitable threshold. To avoid the use of a single method, which can reduce model credibility, two methods were used to determine model thresholds, namely 10 percentile training presence (10th_TP) and maximum training sensitivity plus specificity (MTSS), which were found to be robust in models with only species presence data [51]. MTSS is highly conservative with a high threshold estimation, while 10th_TP is relatively liberal and can complement MTSS [52]. The suitability classes were categorized into non-suitable (<10th_TP), low- (10th_TP~MTSS), moderate- (MTSS~0.6), and high-suitable habitats (≥0.6).

#### 2.3.3. Model Evaluation

AUC and true skill statistic (TSS) were used to assess model accuracy. Higher AUC values indicate more accurate model predictions. AUC is threshold-independent and is used as a common indicator of model performance. The value range of AUC is 0~1 and the model evaluation criteria were poor (AUC ≤ 0.7), fair (0.7 < AUC ≤ 0.8), good (0.8 < AUC ≤ 0.9), and excellent (AUC > 0.9) [53]. TSS is threshold-dependent and is not affected by prevalence and validation set size [54]. TSS ranges from −1 to 1 with a higher value indicating a better prediction. A TSS value > 0.6 indicates an accurate prediction.

## 3. Results

### 3.1. PCA of Climatic Variables

Strong correlations existed among 19 current bioclimatic variables (Appendix A), with *|r|* ≥ 0.75 for 38 pairs of variables. Thus, directly using these variables can drastically affect prediction results. PCA further revealed a total of five principal components with a contribution greater than 0.5% and a cumulative contribution of 99.75%. PC1 contributed 77.10% of the total variability (Table 1), reflecting the synergistic effect of temperature and precipitation. Low PC1 values were associated with seasonal changes in temperature, while high PC1 values suggested a sensitivity to annual and seasonal changes in precipitation (Appendix A). PC2 explained 17.45% of the information in the original variables, mainly indicating seasonal changes in temperature. PC3 and PC4 contributed 3.10% and 1.59%, respectively, and were associated with precipitation changes during the cool and warm seasons, as well as during wet and dry seasons, respectively. High PC5 values (0.52% of explained variance) indicated a high annual precipitation and low PC5 values indicated a low precipitation in cool and warm seasons. The PCA results of bioclimatic variables in 2081–2100 were similar to the current results, leading to the same five principal components and a cumulative contribution of 99.73% of the total variance under both emission scenarios (Table 1).

### 3.2. Niche Differences between Native and Invasive Ranges

Compared to its native niche hypervolume (HV = 140.41), the space of invasive niche of *P. solenopsis* (HV = 1692.14) was one-fold larger, with a low degree of niche overlap (β_total_ = 0.92) between the invasive and native ranges, as is also indicated by the distance between the centroids (1.53). Niche expansion (β_richness_ = 0.91) contributed 99.45% of the niche differentiation, while space replacement (β_replacement_ = 0.01) contributed less than 1%. Niche expansion was extremely pronounced on any of the paired orthogonal axes comprising the five principal components and the native niche hypervolume can be regarded as a subset of the invasive niche hypervolume, in general. Niche expansion occurred mainly along principal component axes other than PC2, favoring high values of PC1, PC3, and PC4, as well as low values of PC5. In contrast, the niche of *P. solenopsis* was conservative on the PC2 axis (Figure 1).

There were significant niche differences between different invasive regions and the native range, with each β_total_ being greater than 0.88 (Table 2). In terms of hypervolume sizes, the niches in different invasive ranges were larger than the native niche. The niche of the Oceania population had the largest hypervolume size of 1917.27, followed by those of Eurasia (1197.40) and Africa (1176.13), which are similar, and the smallest niche was that of South America (963.31). The rank order of centroid distances between the native niche hypervolume and the hypervolumes of different invasive ranges was the same as the rank order of the hypervolume sizes (Table 2).

Compared to the native range of the pest, space replacement of the invasive niche occurred in Africa (β_replacement_ = 0.06), South America (0.05), Eurasia (0.02), and Oceania (0.02) (Table 2 and Figure 2). However, the niche differences were primarily caused by niche expansion with *β*_richness_ of Africa, South America, Eurasia, and Oceania being 0.85, 0.83, 0.88, and 0.92, respectively. Niche expansion contributed more than 90% and was more evident in Oceania and Eurasia. The niches of *P. solenopsis* in different invasive ranges expanded in different directions. The niches in Oceania and Eurasia are expanding to the right on the PC3 axis, while niches in other regions are expanding in both directions. Similarly, the niche in Africa is increasing significantly on the PC4 axis, while that in Eurasia showed a certain degree of contraction (Figure 2).

### 3.3. Model Optimization and Evaluation

The model optimization results showed that among the 48 parameter combinations, there were five parameter combinations with ΔAICc < 2; when FC = LQHP, RM was 0.5, 1, or 3; when FC = LQHPT, RM was 0.5 or 3 (Figure 3). Further comparison of the criterion AUC_diff_ indicated that when FC = LQHPT and RM = 0.5, this led to the lowest value of AUC_diff_, resulting in a model that is able to better differentiate between distribution points and background points and is, therefore, the optimal parameter combination.

The AUC of the model running with the optimal parameters was 0.83 and its TSS was 0.62. This model performed well and can be used for the prediction of suitable habitats for *P. solenopsis.*

### 3.4. Importance of Variables

The Jackknife test showed that among the five principal components, the regularized training gain values were higher with PC1 or PC3 alone, with little difference between the two, followed by PC5 and PC4, and the training gain value was lowest with PC2 alone (Figure 4). The training gain decreased significantly when climatic variables other than PC1 or PC2 were used, suggesting that these two principal components better complemented the training data with more information, which was not present in other variables. In summary, all five principal components were key climatic variables affecting the distribution of *P. solenopsis*.

### 3.5. Potential Distribution in Current and Future Climate Scenarios

Currently, *P. solenopsis* was distributed mainly in eastern and southern Asia, eastern and northern Australia, southern Africa, Central America, south-central South America, and sporadically in Europe and central Africa (Figure 5). Its high-suitable habitats were concentrated in India, Vietnam, Laos, Myanmar, Bangladesh, Pakistan, China, Korea, South Korea, Japan, and Australia. The total suitable distribution area was 3.49 × 10^7^ km^2^, of which the areas of high and medium suitable habitats were 0.62 × 10^7^ km^2^ and 1.33 × 10^7^ km^2^, respectively, accounting for 17.81% and 38.25% of the total area of suitable habitats.

In the SSP1-2.6 scenario for the 2090s, the total area of suitable habitats for *P. solenopsis* would be 4.46 × 10^7^ km^2^, increasing by 27.93% compared to that of current climate conditions (Table 3), with the areas of low-, moderate-, and high-suitable habitats increasing by 17.63%, 33.25%, and 41.93%, respectively. In the SSP5-8.5 scenario, the total area of suitable habitats would increase by 25.82%, with a significant increase in the area of high-suitable habitats (49.25%) and an increase of 18.48% in the area of medium-suitable habitats, which are lower than those in the SSP1-2.6 scenario. Spatially, in the future climate change scenarios, *P. solenopsis* can spread sporadically to North Africa, northern China, areas along the Mediterranean, and northern Europe in a south-to-north spreading pattern, with a significant increase in the area of high-suitable habitats in eastern and southern Asia, especially China, India, and Japan (Figure 6).

## 4. Discussion

*P. solenopsis* is a dangerous pest that has recently invaded ecosystems across the globe. Using PCA and *n*-dimensional hypervolume analysis, this was the first study to explore the multidimensional climatic niche dynamics of *P. solenopsis* in invasive ranges. We built a MaxEnt model based on the optimal parameter combination to assess the global invasion risk for this pest. This study has both theoretical and practical significance, providing scientific support for the early warning and control of *P. solenopsis.*

### 4.1. Niche Shifts

The results of this study showed that the niche differences of *P. solenopsis* between native and invasive ranges was mainly attributed to the niche hypervolume expansion rather than the niche shifts caused by changes in spatial positions (space replacement). The hypervolume size of invasive niches expanded significantly and the expansion direction of niche in different invasive areas was not exactly consistent, probably due to environmental adaptability. Relevant research showed that the evolution of invasive species in their fundamental niches is the core driver for their rapid invasive expansion and that factors such as increased fitness of individuals at the invasion front, different source populations, multiple introductions, and hybridization with closely related species can all accelerate their adaptive evolution [1,21].

Wei et al. [12] used the ordination method to assess the niche conservatism of *P. solenopsis* and found that niche expansion led to climatic niche shifts in Eurasia, which was consistent with our results. However, Wei et al. [12] suggested that niche unfilling led to significant niche shifts in the climatic niches of South America and Oceania, which was inconsistent with the results of this study and is possibly due to two reasons. On the one hand, only niche expansion can be used to characterize niche shifts, and niche unfilling reflects the potential environmental space that the invasive species will subsequently occupy [6,55]. Some scholars even believe that the proportion of niche expansion should be greater than 50% before it can be regarded as a niche shift [56] and our study also found a certain degree of space replacements of climatic niches in South America and Oceania. On the other hand, the ordination method constructs a two-dimensional environmental space, whereas the multidimensional hypervolume analysis used in this study constructs a high-dimensional space. For high-dimensional sparse features or uncorrelated features, the ordination method suffers from a risk of information loss, so it is crucial to increase the cumulative contribution of the first two principal components when quantifying the niche using the ordination method [1,6]. However, the cumulative contributions reported by Wei et al. [12] were not very high, especially in Eurasia, where the number was only 58.6%. Thus, the comparison of multiple methods, such as the ordination approach, the univariate approach, and the hypervolume approach, is recommended to quantify the niche differentiation of *P. solenopsis*, as well as to explore and develop integrated techniques based on these differences.

Despite the large number of studies, there is still significant controversy and a lack of consensus on the niche conservatism of invasive species. A key reason for this discourse lies in the differences of niche quantification methods and the lack of a unified quantification framework [1]. For many species, different studies, and even different analytical methods in the same study, can reach opposing conclusions [3]. For instance, Guo et al. [57] tested the niche conservatism of the globally invasive common reed, *Phragmites australis*, using the ordination, univariate, and ecological niche modeling methods, and found that the validation results were highly dependent on the niche quantification method. In addition, Bates et al. [3] reviewed 135 invasive species niche studies and found that 63.7% of the studies rejected the niche conservatism hypothesis, but also noted that this finding may not be representative, mainly due to the publicized bias favoring niche shift conclusions.

### 4.2. SDM Construction

In this study, the 19 bioclimatic variables were reduced through PCA to five dimensions for simulation analysis. The arbitrary deletion of correlated variables may result in a loss of useful environmental information [58], which may be the reason for the inconsistent results of this study compared to that of Zhang et al. [32]. Zhang et al. [32] directly built the niche hypervolumes using four bioclimatic variables (Bio1, Bio8, Bio10, and Bio17) and the hypervolume size of the climatic niche in the native range was only 40.43. However, there are also shortcomings associated with the use of principal components as prediction variables in MaxEnt. In this study, we showed that all five principal components are key climatic variables affecting the distribution of *P. solenopsis*, but we could not specify the importance of each bioclimatic variable.

When building MaxEnt models, many researchers tend to use the default parameters used by Phillips et al. [59] in their early tests on 266 species of birds, mammals, reptiles, etc., with the aim of predicting realistic species distributions. It has been found that MaxEnt models based on complex mechanical algorithms are sensitive to sampling bias and simply using default parameters can lead to overfitting, making the results of predicting potential distributions unreliable and sometimes uninterpretable [17,18,49]. In this study, we used ENMeval 2.0 to optimize the FC and RM parameters of the model and found that the model based on the default parameters had a higher AUC_diff_, indicating the necessity for model optimization, which had not been taken into account in previous predictions of suitable habitats for *P. solenopsis* [12].

In recent years, SDMs have been widely used in invasive biology research [5,21,56]. Niche shifts reduce the transferability of an SDM, which is particularly evident during biological invasions [2]. In this study, we found that *P. solenopsis* had undergone a significant niche expansion in its invasive ranges and using an SDM based on its global distribution to predict its global suitable habitats poses certain risks. Niche expansion in invasive ranges may lead to the development of unique niche characteristics in different invasive ranges and, therefore, limiting the scope of simulated areas is recommended.

### 4.3. Invasion Risks and Management Recommendations

In 2017, Wei et al. [12] analyzed the global suitable habitats for *P. solenopsis* and categorized its high-suitable habitats using the same threshold as we used in this study. The area of the high-suitable habitats predicted in 2017 is 0.28 × 10^7^ km^2^, whereas the area predicted by this study is 0.62 × 10^7^ km^2^, which is consistent with the rapid invasion of *P. solenopsis* in recent years. Since 2017, *P. solenopsis* has been found for the first time in North Ethiopia [35], Algeria [37], Shandong in northern China [33], Saudi Arabia [41], Italy [38], Kenya [39], Morocco [40], and Tunisia [42], indicating the startling reality of the northwards invasion of this species. This pest is extremely adaptable to temperature, being especially tolerant to high temperatures [60], and its habitats are likely to expand significantly in the future under climate changes, with North Africa, northern China, areas along the Mediterranean, and northern Europe being high-risk areas for invasion.

Compared to natural spreading, the management of the unintentional introduction of infected plants is key to *P. solenopsis* control [21,22]. *P. solenopsis* has a limited ability to move autonomously and anthropogenic factors are the main pathway for their long-distance spreading. *P. solenopsis* is small in size and not easily detected during the transportation of the host plants. The modern logistics and flower industry have significantly contributed to the spreading of *P. solenopsis* across geographical barriers. Plants, which are kept over an extended period in the flower market, and areas surrounding the logistics center can provide suitable environmental conditions for *P. solenopsis* populations [22]. In areas with high invasion risks, extra attention should be paid to the quarantine of host plants such as cotton, fruits, vegetables, and flowers.

## Figures and Tables

**Figure 1 insects-15-00250-f001:**
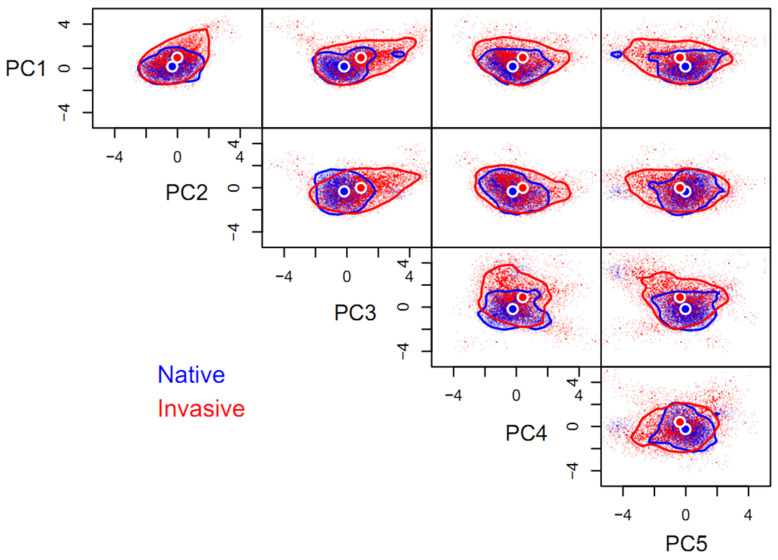
Five-dimensional hypervolume sizes of *Phenacoccus solenopsis* in its native and invasive ranges. Circles represent centroids of each hypervolume. In total, 10,000 random points were sampled from each hypervolume to delineate its shape and boundary. The same applies below.

**Figure 2 insects-15-00250-f002:**
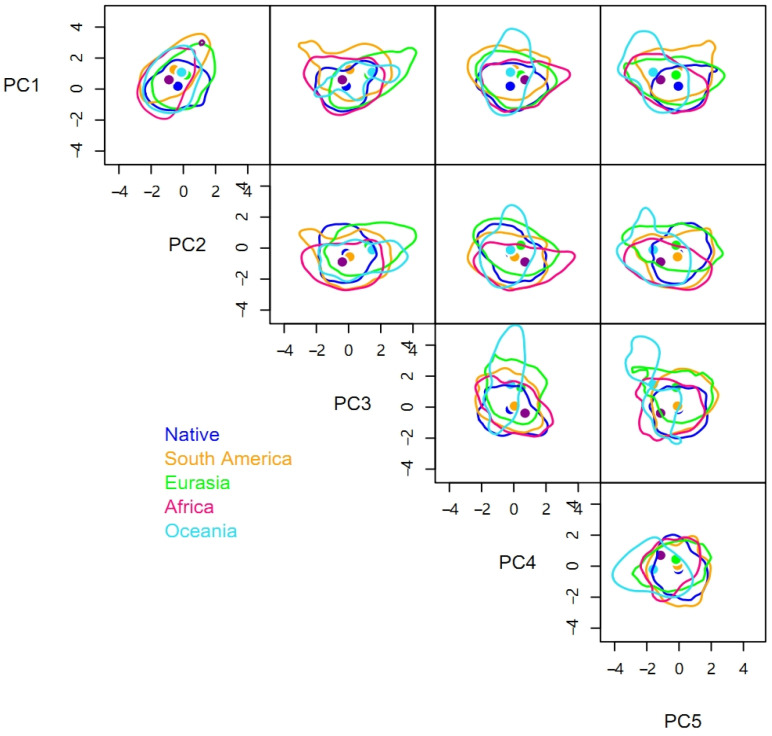
Five-dimensional hypervolume sizes of *Phenacoccus solenopsis* in its native and different invasive ranges.

**Figure 3 insects-15-00250-f003:**
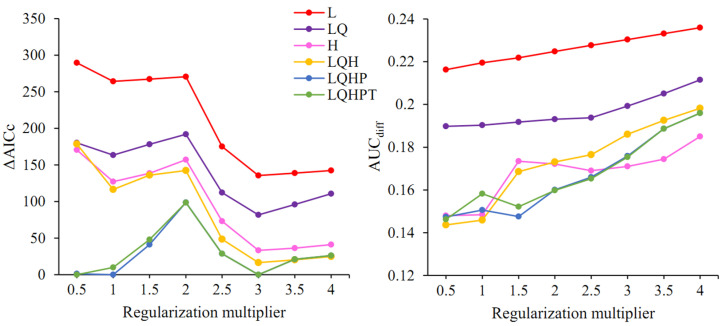
ΔAICc and AUC_diff_ values of the MaxEnt model under different setting parameters. L: linear; H: hinge; Q: quadratic; P: product; T: threshold.

**Figure 4 insects-15-00250-f004:**
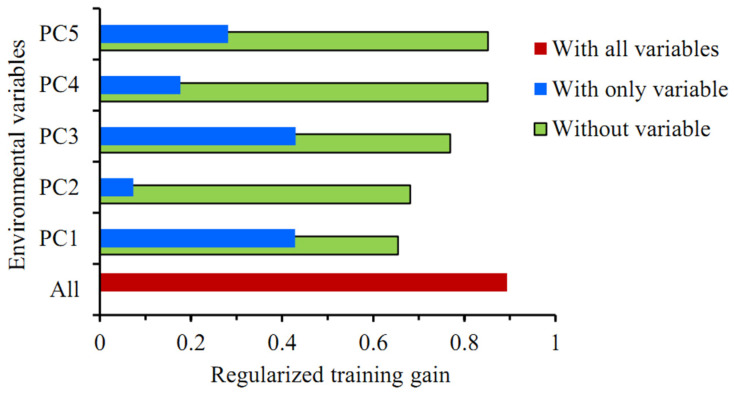
The Jackknife test for evaluating the relative importance of environmental variables in the MaxEnt model.

**Figure 5 insects-15-00250-f005:**
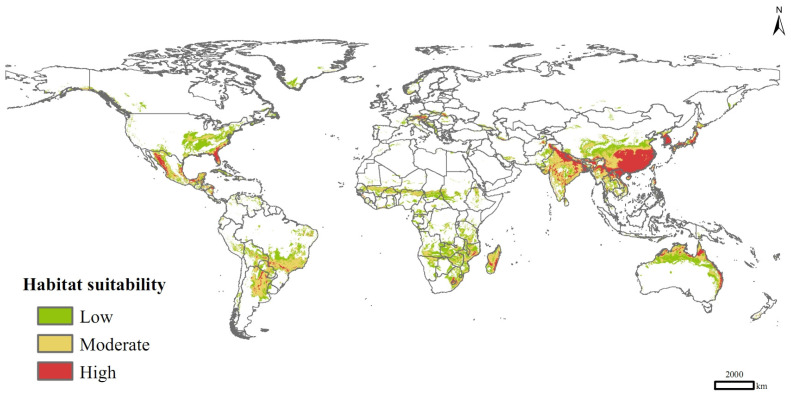
Potential distribution of *P. solenopsis* based on current climate variables.

**Figure 6 insects-15-00250-f006:**
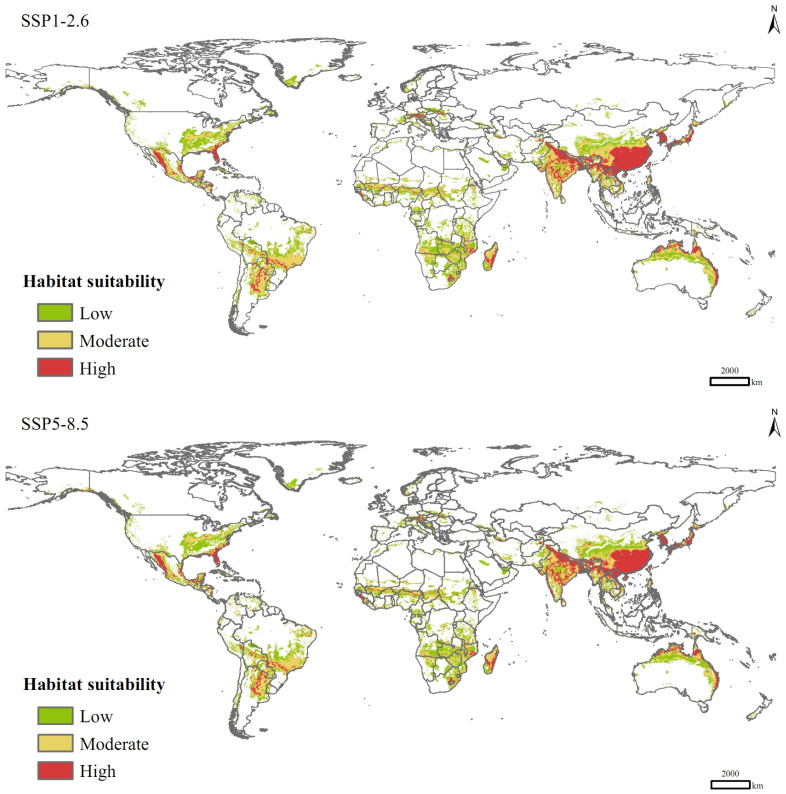
Potential distribution of *P. solenopsis* under different compulsion scenarios in 2081–2100.

**Table 1 insects-15-00250-t001:** Percentage contribution of first five principal components in PCA under current and future climate scenarios.

PC	Current	2081–2100	
		SSP1-2.6	SSP5-8.5
PC1	77.10	76.89	76.48
PC2	17.45	16.81	17.16
PC3	3.10	3.40	3.44
PC4	1.59	2.00	1.99
PC5	0.52	0.63	0.65
Cumulative contribution	99.75	99.73	99.73

**Table 2 insects-15-00250-t002:** Niche differentiation among 5-dimensional hypervolumes of the native and invasive populations of *P. solenopsis*.

	Native Range	South America	Eurasia	Africa	Oceania
Native range	—	1.19	1.79	1.76	2.54
South America	0.88 = 0.05 + 0.83	—	1.46	1.63	2.21
Eurasia	0.89 = 0.02 + 0.88	0.67 = 0.52 + 0.15	—	2.27	1.63
Africa	0.91 = 0.06 + 0.85	0.58 = 0.44 + 0.14	0.69 = 0.68 + 0.01	—	2.40
Oceania	0.94 = 0.02 + 0.92	0.75 = 0.33 + 0.42	0.71 = 0.41 + 0.30	0.64 = 0.32 + 0.32	—
Hypervolume	140.41	963.31	1197.40	1176.13	1917.27

Above the diagonal was the centroid distance, below the diagonal was the niche overlap (β_total_ = β_replacement_ + β_richness_).

**Table 3 insects-15-00250-t003:** Areas of suitable habitats for *P. solenopsis* under different climate scenarios (×10^4^ km^2^).

Habitat Suitability	Current	2081–2100	
		SSP1-2.6	SSP5-8.5
Low	1530	1800 (17.63%)	1878 (22.70%)
Moderate	1333	1777 (33.25%)	1580 (18.48%)
High	621	881 (41.93%)	927 (49.25%)
Total	3485	4458 (27.93%)	4384 (25.82%)

The percentage in brackets indicates the ratio of change in the area of suitable habitats in the future climate scenario compared to the current climate.

## Data Availability

The data presented in this study are available on reasonable request from the corresponding author.

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
