# Peer review of "Hypervolume Niche Dynamics and Global Invasion Risk of Phenacoccus solenopsis under Climate Change"

_insects, 2024, doi:10.3390/insects15040250_

Round 1

Reviewer 1 Report

Comments and Suggestions for Authors

The manuscript concerns an important topic: biological invasion. Climate change increases the range of invasive species to which Phenacoccus solenopsis belongs.

Predicting the ranges of invasive species is an important research problem enabling understanding the threats caused by biological invasions.

The manuscript is interesting, but in my opinion some additions are necessary.

It is worth emphasizing that authors clearly explain the differences in known approaches for analysing niche dynamics.

I suggest adding the definition of invasive species to Introduction. Sometimes there are discrepancies in the definition of the invasiveness criterion of a species, so it is important that the authors provide definition.

Information from the online database Scalenet should be cited: García Morales M, Denno BD, Miller DR, Miller GL, Ben-Dov Y, Hardy NB. 2016. ScaleNet: A literature-based model of scale insect biology and systematics. Database. doi: 10.1093/database/bav118. http://scalenet.info.

It is necessary to add information about the number of generations of Phenacoccus solenopsis per year, because it is important in the case of invasive species.

More information about the occurrence of Phenacoccus solenopsis in other parts of the world including Europe and America should be added.

I think, that authors should explain if the distribution points mentioned in literature differ in each publication or whether they overlap at least partially.

Line 121: I wonder if the distribution points used for analysis were selected  from these 706 points mentioned above. Please, explain this.

Figure 2: The two colours “native” and “Africa” are very similar, it will be better to use clearly different colours.

Reviewer 2 Report

Comments and Suggestions for Authors

This paper is already in great shape.  I have only a couple of minor things to fix in the methods:

Need to cite R and R packages

Line 165: you underline the name of the function, but you don’t do this elsewhere.  You’ve used italics for package names; hoose a consistent way to write function names.

Line 175 and elsewhere: you should mention that of the various SDM paradigms, MAXENT was specifically developed for use with presence-only data.

Author Response

Thank you very much for your kind help and insightful suggestions on our manuscript.We have provided the comments here in bold. Our responses follow these and are highlighted in blue.

 -This paper is already in great shape.

We deeply appreciate your positive comments.

 -I have only a couple of minor things to fix in the methods:

Need to cite R and R packages

 Thanks a lot for the reminding. We have added the citations in the revised manuscript.

-Line 165: you underline the name of the function, but you don’t do this elsewhere.  You’ve used italics for package names; choose a consistent way to write function names.

 Thank you very much for your careful correction. We have deleted all underlines of the function names.

-Line 175 and elsewhere: you should mention that of the various SDM paradigms, MAXENT was specifically developed for use with presence-only data.

 As suggested, we have mentioned the strength of MaxEnt in line 180.

Reviewer 3 Report

Comments and Suggestions for Authors

The authors present an interesting study informing on the important field of model prediction for informing on potential species range change-especially expansion. This is to some extent a contentious issue without consensus as to ‘best’ model. Consideration of outcomes with different modelling approaches is useful beyond interest in the specific pest (cotton mealybug) under consideration here-thus the authors provide a well written and well analysed study with both theoretical and practical implications.

I have a single comment – to insert a common name for or descriptor to indicate the type of organism P. australis (L369)

Author Response

Thank you very much for the careful review. We have provided the comments here in bold. Our responses follow these and are highlighted in blue.

-The authors present an interesting study informing on the important field of model prediction for informing on potential species range change-especially expansion. This is to some extent a contentious issue without consensus as to ‘best’ model. Consideration of outcomes with different modelling approaches is useful beyond interest in the specific pest (cotton mealybug) under consideration here-thus the authors provide a well written and well analysed study with both theoretical and practical implications.

We deeply appreciate your positive comments.

 -I have a single comment – to insert a common name for or descriptor to indicate the type of organism P. australis (L369)

 Thanks a lot for the reminding. We have added a common name of P. australis: "the globally invasive common reed Phragmites australis…." in line 372.